

# Soundscape manipulation enhances larval recruitment of a reef-building mollusk

Ashlee Lillis*, DelWayne R. Bohnenstiehl and David B. Eggleston

Department of Marine, Earth & Atmospheric Sciences, Center for Marine Sciences & Technology, North Carolina State University, Raleigh, NC, USA
* Current affiliation: Woods Hole Oceanographic Institution, Woods Hole, MA, USA

## ABSTRACT

Marine seafloor ecosystems, and efforts to restore them, depend critically on the influx and settlement of larvae following their pelagic dispersal period. Larval dispersal and settlement patterns are driven by a combination of physical oceanography and behavioral responses of larvae to a suite of sensory cues both in the water column and at settlement sites. There is growing evidence that the biological and physical sounds associated with adult habitats (i.e., the "soundscape") influence larval settlement and habitat selection; however, the significance of acoustic cues is rarely tested. Here we show in a field experiment that the free-swimming larvae of an estuarine invertebrate, the eastern oyster, respond to the addition of replayed habitat-related sounds. Oyster larval recruitment was significantly higher on larval collectors exposed to oyster reef sounds compared to no-sound controls. These results provide the first field evidence that soundscape cues may attract the larval settlers of a reef-building estuarine invertebrate.

## INTRODUCTION

Most marine benthic communities are established and maintained via the settlement of larvae, following the development and dispersal of planktonic early life stages of fish and invertebrates. Larval habitat selection and settlement to suitable juvenile and adult habitat is critical for subsequent survival and reproductive success, and ultimately shapes species distributions and their population dynamics (*Gaines & Roughgarden, 1985*; *Caley et al., 1996*). Locating favorable settlement sites in a vast ocean environment following days to months in the water column presents a significant challenge. While most larvae are relatively weak swimmers compared to the speed at which currents transport them, they use a complex suite of cues to encounter and select settlement habitat (*Rittschof et al., 1998*; *Kingsford et al., 2002*). Larval orientation and settlement cues include changes in salinity, tidal direction and turbulence that can aid larvae trying to move from oceanic to estuarine environments, as well as odors and bacterial films that help larvae settle on advantageous substrates and in sufficient proximity to one another to facilitate future reproductive success as adults (*Rittschof et al., 1998*; *Forward, Tankersley & Rittschof, 2001*). Patchy

Corresponding author
Ashlee Lillis, ashleelillis@gmail.com

marine ecosystems such as reefs, seamounts and deep-sea vents may be especially difficult to encounter by larvae without broad-scale cues. Physical and chemical characteristics of the water column and seafloor (e.g., flow, light, habitat odors, texture) can influence larval swimming and settlement behaviors at multiple scales (*Bourget, 1988*; *Butman, Grassle & Webb, 1988*; *Kingsford et al., 2002*), and settlement outcomes are likely the result of complicated interactions of larvae with environmental variables; however, their effect on larval settlement in the field is seldom directly measured.

An emerging area of research in marine ecology is the role that underwater sounds play as a cue for larval orientation and settlement of fishes and invertebrates. Habitat-related soundscapes (the combination of sounds forming an immersive acoustic environment at a particular location) may represent a valuable cue for a variety of larvae because these underwater sounds can indicate both the presence and biophysical characteristics of particular habitat types, and sounds can travel independent of currents over greater distances compared to other habitat cues (e.g., chemical odor) (*Montgomery et al., 2006*; *Cotter, 2008*; *Lillis, Eggleston & Bohnenstiehl, 2014a*; *Lillis, Eggleston & Bohnenstiehl, 2014b*). Recent studies have established underwater sound as an orientation and settlement cue for a variety of fish and crustacean larvae, particularly in coral and rocky reef systems (*Tolimieri, Jeffs & Montgomery, 2000*; *Simpson et al., 2005*; *Stanley, Radford & Jeffs, 2012*). However, this phenomenon has not been tested in estuarine ecosystems, where high habitat diversity supports a large array of commercially and ecologically important species producing planktonic larvae. Because of their ecological role as a habitat-creating species in estuaries, as well as their economic importance and global demise, we sought to test whether the sounds of oyster reefs could enhance recruitment of larval oysters (*Crassostrea virginica*). The specific sensory mechanism by which invertebrate larvae may detect acoustic stimuli has not been determined, but late-stage larval oysters possess both exterior cilia and statocyst sensory structures (*Kennedy, Newell & Eble, 1996*), which have been shown to be responsive to acoustic particle motion in other aquatic invertebrates (*Rogers & Cox, 1988*; *Budelmann, 1992*; *Zhadan, 2005*).

Populations of reef-building bivalve molluscs of the family Ostreidae create important intertidal and subtidal biogenic habitats throughout temperate estuarine and coastal ecosystems worldwide (*Gutiérrez et al., 2003*; *Beck et al., 2011*; *Maslo, 2014*), including the Atlantic and Gulf coastlines of North America where oyster reefs were once vast prominent features (*Jackson et al., 2001*). Reef-building organisms such as these generate conspicuous habitat that is vital in providing shelter for numerous associated fish and invertebrates (high local abundance and diversity) and also in carrying out a range of ecosystem services in these highly exploited and degraded systems, such as enhancing benthic-pelagic coupling and nutrient cycling via filtration of large amounts of material from the water column, as well as sequestering carbon in their shells as they grow (*Dame, Zingmark & Haskin, 1984*; *Maslo, 2014*). Moreover, oysters are an economically valuable resource, but native populations are now less than 5% of their historical abundances due to fishing pressure, oyster bed destruction, habitat degradation and disease (*Jackson et al., 2001*; *Beck et al., 2011*). Ecosystem restoration efforts to recover the economic and ecological benefits

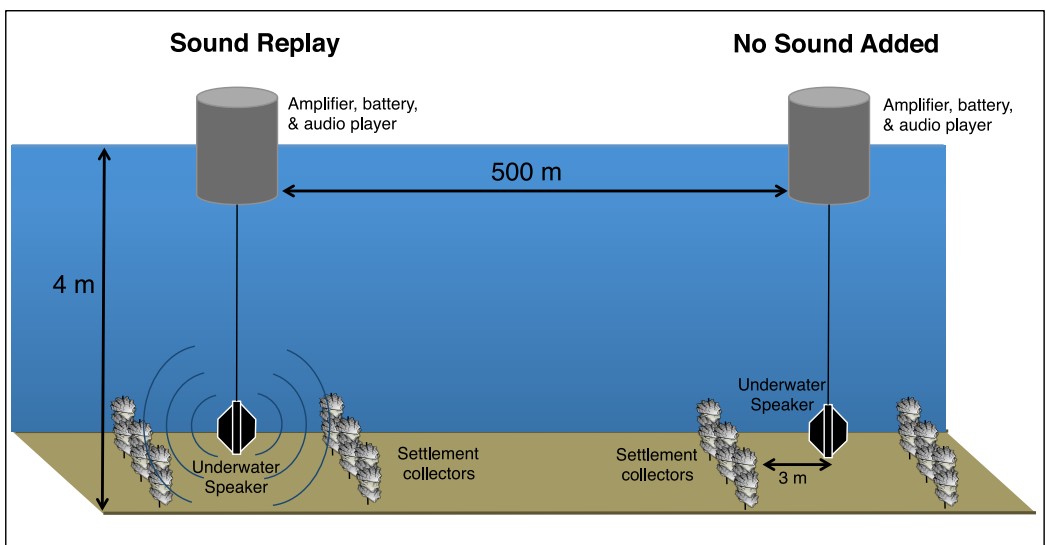

**Figure 1 Experimental setup schematic.** Design of the experimental sites, showing a pair of underwater speaker deployments separated by 500 m, producing two distinct soundscape treatments. Larval collectors, constructed of non-living oyster shells, were arranged around each speaker unit.

of oyster reef habitat have become common; however, restoration failures have highlighted the need to better understand oyster life history and previously unconsidered aspects of their biology (*Geraldi et al., 2013*; *Maslo, 2014*).

We recently reported that the soundscapes of subtidal oyster reefs in Pamlico Sound, North Carolina, USA have distinct acoustic properties compared to surrounding soft mud bottoms, with reef soundscapes comprised of higher levels of sound across frequencies produced by soniferous reef-dwellers such as snapping shrimp and oyster toadfish (*Lillis, Eggleston & Bohnenstiehl, 2014a*). Additionally, experiments using cultured oyster larvae indicate that oyster reef sound can increase settlement in small experimental chambers, suggesting that a larval response to reef sound could facilitate encounter and influence oyster recruitment patterns (*Lillis, Eggleston & Bohnenstiehl, 2013*; *Lillis, Eggleston & Bohnenstiehl, 2014b*). To advance our understanding of the ecological significance of the observed larval settlement response to sound by oyster larvae, the present study tests if replayed sounds from preferred adult habitat influence recruitment of free-swimming larvae in a natural setting.

## MATERIALS AND METHODS

We measured the effect of replayed oyster reef sound on oyster larval recruitment by comparing the density of newly settled oysters (termed "spat") on collectors deployed in a mud flat area with or without added recorded reef sound. Two experimental areas were constructed at a field site in West Bay, Pamlico Sound, NC. Submersible speakers were deployed with spat collectors placed 3 m from the speaker (Fig. 1). Eight trials lasting 3–5 days were conducted during July and August, alternating the sound treatment between the two experimental areas. For each trial, the sound replay treatment used recordings taken

from an oyster reef site more than 2 km away. Permission to conduct fieldwork in Pamlico Sound waters and the West Bay oyster reserve was granted by the North Carolina Division of Marine Fisheries permit numbers 708396 and 1012889.

## Spat collectors and site design

Collectors for larval oyster settlers were constructed of 10 cleaned adult oyster shells (their preferred settlement substrate) strung on a ~15 cm piece of wire. Each replicate came from five or six collectors located >1 m apart and 3 m from a speaker, suspended ~1 m from the seabed (Fig. 1). To avoid inter-site differences in larval delivery both experimental sites were positioned parallel to the axis of the dominant current direction in the bay, situated 500 m apart and equidistant from the West Bay oyster reserve (>2 km away) (Fig. 1). The location was selected to allow the no sound control treatment to be as representative of off-reef soft-bottom soundscape as possible. The 500 m site separation was also intended to minimize interference from the sound replay treatment at the control site, while still exposing treatments to the same larval pool. During previous acoustic surveys, elevated sound levels associated with reef environments were found to diminish by 15–20 dB and be spectrally similar to soft-bottom habitat soundscapes at 500 m from reefs (*Lillis, Eggleston & Bohnenstiehl, 2014a*, see Fig. 8).

## Sound treatments

A library of recordings collected in July & August of 2010 and 2011 were used as the sound broadcast for the "sound added" treatments. A different recording was randomly selected for each of the eight trials, and consisted of 15 one-minute files from the West Bay oyster reserve, looped continuously to provide a constant "oyster reef" replay. Sound replay systems consisted of a submersible speaker (LL916; Lubell Labs, Columbus, Ohio; frequency response: 0.2–20 kHz) connected to a surface buoy containing a power amplifier (Peavey IPA 1502; Peavey, Meridian, Mississippi, USA), using a handheld digital media player (Apple iPod; Apple, Cupertino, California, USA) as audio input, and powered by a 12 V battery.

The acoustic conditions during trials were monitored using DSG acoustic recorders (Loggerhead Instruments, Sarasota, Florida, USA), each equipped with an HTI-96 hydrophone (High-Tech Inc., Gulfport, Mississippi, USA) with a flat frequency response between ~0.1 and 30 kHz. The DSG instruments digitize acoustic data using a 16-bit resolution written to a standard solid-state SD memory card. A recorder was deployed with spat collectors 3 m from the speaker at each experimental site, with the hydrophone sensor positioned at 1 m from the seabed, and programmed to record for 1 min at 10 min intervals for the duration of a trial (sampling rate = 50 kHz). Acoustic data are available to characterize the soundscape generated at the sound replay sites for all trials except trial 4, but due to instrument unavailability or malfunction, acoustic monitoring was not possible at the no sound site for all trials (see Supplemental Information).

Analysis of recordings collected during trials showed that the average root-mean-square broadband sound pressure level recorded at the site with replayed sound treatment ranged between 125.4–135.9 dB re 1 μPa, compared to 110.3–115.9 dB re 1 μPa recorded

at the control site with no replayed sound. Replayed reef sound recordings primarily consisted of snapping shrimp and oyster toadfish calls, differing from the no sound control treatment acoustic composition largely in frequencies >800 Hz (Figs. 2B and 2C). Site recordings during trials confirmed that the acoustic treatments were distinct in frequency composition (Figs. 2B and 2C), and that the relative acoustic spectra were similarly shaped to the original reef and off-reef soundscapes (Fig. 2A), although sound levels in upper frequencies (>1 kHz) were not as low at the "no sound" treatment sites as previously measured for off-reef soft-bottom habitat. This could be the result of a small influence of the nearby sound replay treatment on the entire experimental area, temporal differences in the off-reef soundscape compared to the recordings from the previous year, or additional sound sources at the site. Nonetheless, the acoustic monitoring during trials demonstrates that the replayed sound treatment was effective in substantially increasing sound levels in reef-associated frequencies, and producing a distinct soundscape at the manipulated treatment sites.

## Statistical analysis

We tested the null hypothesis that the number of recruits on collectors exposed to replayed sound will be the same as the number on collectors not exposed to replayed sound, using an exact binomial test, with an expected proportion of 0.5, to estimate the probability of observing $k$ or more recruits on the replayed sound collectors given $n$ total recruits during an experimental trial:

$$p(\geq k, n, 0.5) = 1 - \sum_{j=0}^{k-1} \binom{n}{j} 0.5^j (1 - 0.5)^{n-j}$$

where $\binom{n}{j} = \frac{n!}{j!(n-j)!}$

# RESULTS

Oyster settlement varies throughout the summer reproductive period, and therefore there was a significant effect of trial date on total recruitment, as the majority of settlement during the study period occurred during the latter part of August (Fig. 3). A significantly higher proportion of oyster recruits were found at the replayed sound site for the each of the first six out of eight trials (Fig. 3). In these trials, between 68.3–100% of larvae recruited on collectors at sound treatment sites (Fig. 3 and Table 1). There was no difference in recruitment between treatments in the two late August trials, which were conducted during a period of peak larval settlement (Fig. 3). Pooled over the duration of the experiments, collectors exposed to replayed sound received 58.4% of the total oyster recruitment ($p < 0.0001$, $n = 1,685$).

The first six experiments show an average of 83% of the recruitment occurring on the sound replay collectors. These differences are significant with $p(\geq k, n, 0.5)$ ranging from <0.0001 to 0.0384 for the individual experimental trails. The last two experiments, which exhibit much higher overall recruitment rates, show slightly fewer recruits on the sound replay collectors; however, these differences are not significant (Table 1).

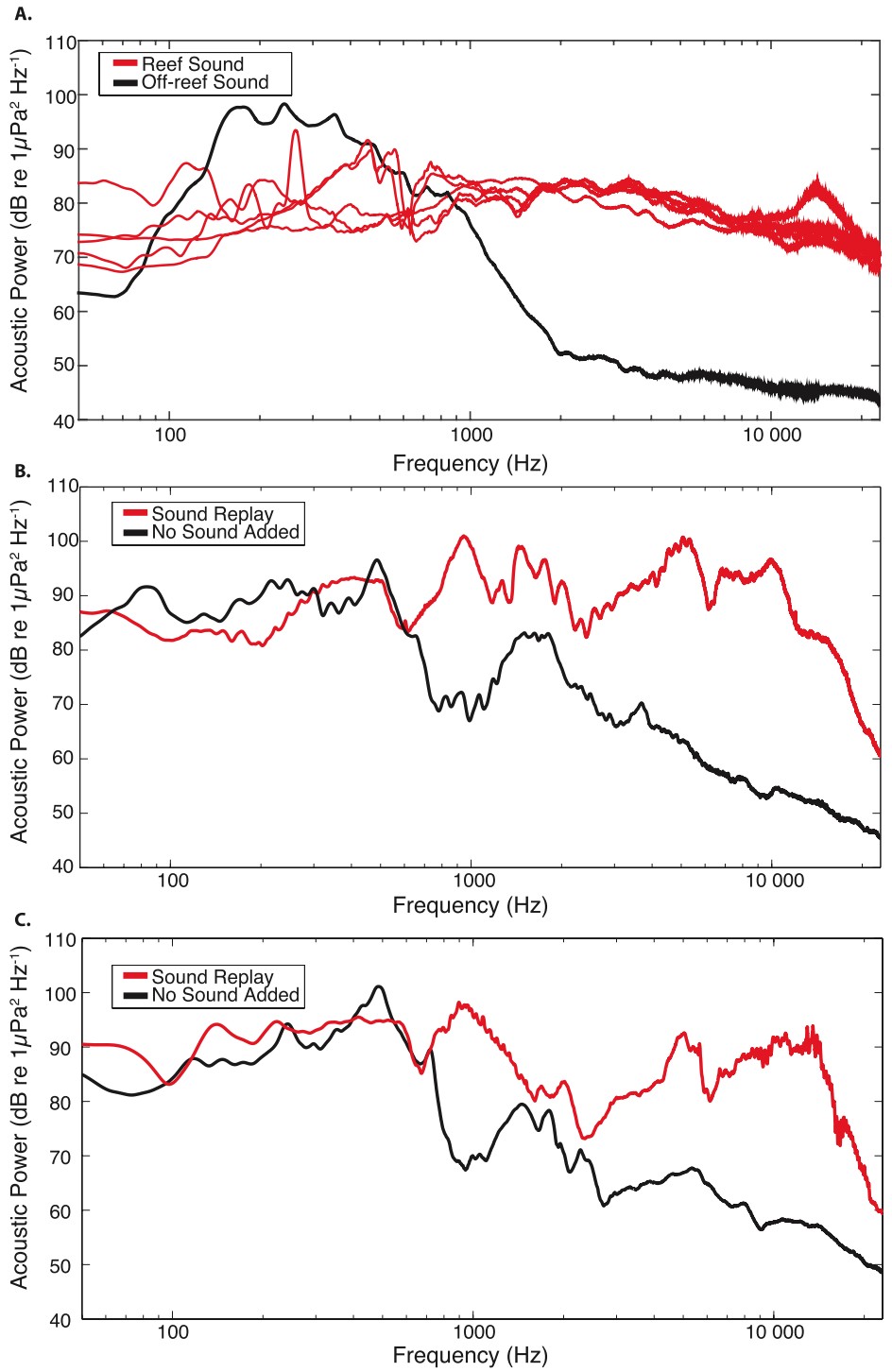

**Figure 2  Acoustic power spectra for original and replayed reef recordings, compared to off-reef and "no sound" control recordings.** (A) Power spectral analysis of original 15 min recordings used as sound replay treatments from July & August 2011. For comparison, the black line represents the acoustic spectrum for a 15 min recording made in off-reef soft-bottom habitat during July 2011. Power spectral densities were estimated using the median spectra obtained from a series 0.5 s 

**Figure 2 (...continued)**
duration non-overlapping Hanning-windowed data segments spanning each 15 min recording (FFT size: 32,768 points, frequency resolution: 1.5 Hz). Acoustic conditions recorded at the two treatment sites during the experimental field replay are shown for two trials in (B) 7/13–7/16 and (C) 8/23–8/27. Red lines represent the recorded replayed sound at the collector sites; Black lines represent ambient sound level during the same time period at the control (no sound) collector sites. The median power spectral densities for treatments in (B) and (C) were generated from a series of non-overlapping 0.5 s duration Hanning-windowed data segments spanning each 1 min recording.

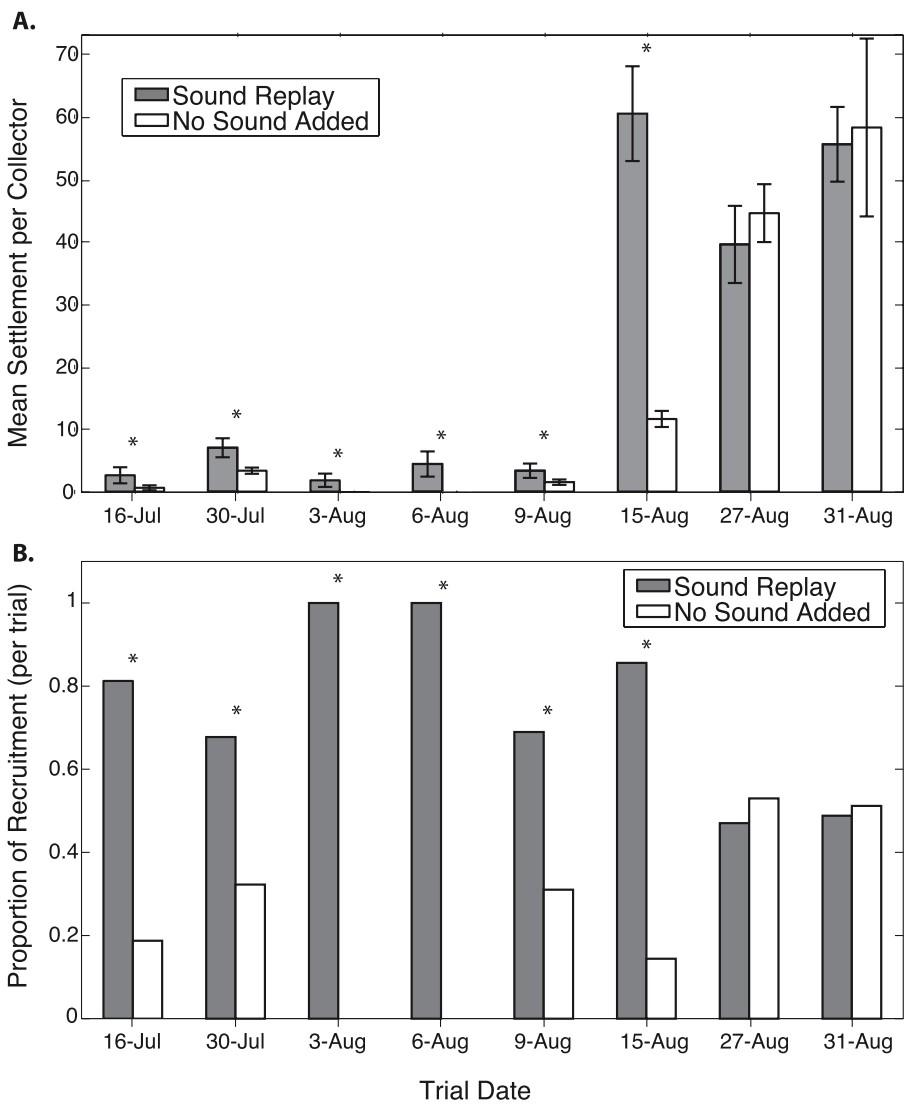

**Figure 3** **Oyster recruitment on spat collectors exposed to different acoustic treatments.** Comparison of recruitment on collectors exposed to reef sound replay and collectors with no added sound, in each of eight trials. Shown as (A) mean recruitment per collector (±1 S.E.), and (B) proportional recruitment between the two treatments in each trial.

**Table 1  Summarized experimental trial information and results.**

| Trial dates (2012) | Trial length (hours) | No. collectors per treatment | Total spat count $n$ | % Spat on collectors with sound replay ($k$) | $p(\geq k, n, 0.5)$ [*] |
|---|---|---|---|---|---|
| 13–16 Jul | 72 | 5 | 16 | 75.7 (12) | 0.0384 |
| 26–30 Jul | 96 | 6 | 63 | 68.3 (43) | 0.0026 |
| 30 Jul–3 Aug | 96 | 5 | 9 | 100.0 (9) | 0.0020 |
| 3–6 Aug | 72 | 5 | 22 | 100.0 (22) | <0.0001 |
| 6–9 Aug | 72 | 6 | 29 | 69.0 (20) | 0.0307 |
| 10–15 Aug | 120 | 5 | 360 | 85.6 (308) | <0.0001 |
| 23–27 Aug | 96 | 6 | 504 | 47.0 (237) | 0.9164 |
| 27–31 Aug | 96 | 6 | 682 | 48.8 (333) | 0.7171 |

Notes.

[*] Probability of observing $k$ or more spat on the replayed sound collectors, given $n$ total recruits and an expected proportion equal to 0.5. For pooled data: $p(\geq k, n, 0.5) < 0.0001$, $k = 984$, $n = 1,685$.

## DISCUSSION

This study provides field evidence that sound influences settling oyster larvae and is the first to indicate that an estuarine invertebrate actively selects substrate associated with the acoustic cues from its preferred adult habitat. These results demonstrate that protection or enhancement of reef soundscape characteristics should be included in the management and restoration of this important estuarine reef-building species, and that sounds from oyster reefs may be used to enhance larval settlement in oyster hatchery operations that support aquaculture. The use of chemical cues, biofilms and attractive substrates (*Alfaro et al., 2006*; *Roberts & Watts, 2010*; *Li et al., 2014*) are commonly incorporated in aquaculture and restoration settings to promote larval settlement. The results of the experiments reported here suggest that soundscape characteristics should additionally be considered in endeavors to optimize larval settlement conditions. In our experiments, larval recruitment was higher on collectors in reef sound replay treatments compared to no added sound, but further trials using a positive control (i.e., white noise played at the same sound level as the reef sound replay) will be needed to fully assess whether larvae respond specifically to habitat-related sound or if general elevated sound levels induce a response.

Field experiments in which the densities of newly settled attached organisms are measured unavoidably integrate patterns of both larval settlement and early post-settlement processes, since some amount of juvenile mortality will occur before observations can be made (*Keough & Downes, 1982*). In this experiment, it is therefore not possible to confirm that the differences between recruitment on sound versus no sound spat collectors were due to larval settlement processes. However, given the relatively short trial periods (3–5 days), and prior results showing elevated settlement to soundscape treatments in larval cultures (*Lillis, Eggleston & Bohnenstiehl, 2013*), it is most likely that the recruitment differences are the result of habitat selection and active larval response to the replayed sound. Moreover, if the acoustic manipulation was predicted to have an effect on post-settlement mortality, we would anticipate opposite or null results since the oyster reef sound replay would be expected to attract more predators than the no

sound added control. While this study measured effects of elevated sound on initial recruitment only, and further study is required to test if the addition of reef sound could lead to long-term differences in oyster density, previous studies have found that annual juvenile oyster recruitment correlates with spatial patterns of spatfall, despite high levels of post-settlement mortality (*Newell et al., 2000*). Thus, we would predict that differences in spat density produced by soundscape variation could contribute to larger scale patterns of oyster abundance and distribution.

The lack of a statistically significant oyster recruitment response in the last two of eight field trials may have been due to very high larval availability during August when there is a peak in oyster spat settlement in Pamlico Sound (*Eggleston et al., 2011*), such that high recruitment densities swamped any sound treatment effect. The Pamlico Sound estuarine system is substrate limited for oysters (*Geraldi et al., 2013*), so it is unsurprising that larval settlement occurs with less selectivity under high larval supply. At higher densities more individuals in a population will occupy sub-optimal habitat (*Rosenzweig, 1991*) and in this experiment we found that recruitment patterns do not relate to the acoustic conditions when the larval pool is at its largest. In previous studies, substrate related chemical cues were found to be less influential on settlement patterns for barnacle larvae settling later in the reproductive season (*Jarrett, 1997*), and larval sensitivity to cues is known to decrease over the larval period for many species (*Gibson, 1995*; *Elkin & Marshall, 2007*). Despite the lack of significant effect of replayed sound on recruitment in the final two trials, the overall effect of sound was statistically significant, and our results suggest that the influence of an acoustic settlement cue might be particularly significant under conditions where larvae are limited.

Given the significance of habitat selection in the lifecycle of oysters and other reef-building organisms, and the unprecedented threats these ecosystems currently face, understanding the drivers of the settlement process is key to successful prediction of population dynamics, and accurate biophysical models of larval recruitment. This study reveals a previously unrecognized effect of the soundscape on a key ecological process for an ecosystem engineer. Establishing the influence of sounds on the early stages of weakly swimming reef-building organisms has broad implications for marine ecology, including marine conservation and aquaculture programs, and underscores the importance of the ambient acoustic environment as a landscape-scale structuring component of benthic ecosystems.

## ACKNOWLEDGEMENTS

We are grateful to J Peters and R Dunn for assistance in field experiment development and execution, to B Puckett for additional field assistance, and to D McVeigh and S Brown for lab sample processing. Thank you to J Luczkovich and D Kamykowski, who gave feedback on the initial design of this study.

### Funding

Funding was provided by the National Science Foundation (Grants OCE-1234688 & ISO-1210292). Additional support for experimental materials came from a PADI Foundation Grant (#5145) and a National Shellfisheries Association Melbourne R. Carriker Student Research Grant to AL. The funders had no role in study design, data collection and analysis, decision to publish, or preparation of the manuscript.

### Grant Disclosures

The following grant information was disclosed by the authors:
National Science Foundation: OCE-1234688, ISO-1210292.
PADI Foundation Grant: #5145.
National Shellfisheries Association Melbourne R. Carriker Student Research Grant.

### Competing Interests

The authors declare there are no competing interests.

### Author Contributions

- Ashlee Lillis conceived and designed the experiments, performed the experiments, analyzed the data, contributed reagents/materials/analysis tools, wrote the paper, prepared figures and/or tables.
- DelWayne R. Bohnenstiehl conceived and designed the experiments, analyzed the data, contributed reagents/materials/analysis tools, reviewed drafts of the paper.
- David B. Eggleston conceived and designed the experiments, contributed reagents/materials/analysis tools, reviewed drafts of the paper.

### Field Study Permissions

The following information was supplied relating to field study approvals (i.e., approving body and any reference numbers):

Permission to conduct field work in Pamlico Sound waters and the West Bay oyster reserve was granted by the North Carolina Division of Marine Fisheries permit numbers 708396 and 1012889.

### Supplemental Information

Supplemental information for this article can be found online at http://dx.doi.org/10.7717/peerj.999#supplemental-information.

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
