# Peer review of "Soundscape manipulation enhances larval recruitment of a reef-building mollusk"

_PeerJ, doi:10.7717/peerj.999_

## Round 0.1 · original submission · Minor Revisions

Dear Author,

Thank you for submitting your manuscript to PeerJ. Please note that one of the reviewers has made a couple of suggestions to improve the manuscript you may want to consider.

·

Basic reporting

This is a timely, well-written ms that elegantly highlights a rather underappreciated field in marine ecology. The authors have done a good job conveying this point using a very simple, but effective experimental design. The Discussion nicely elaborates on the authors' findings and provides the appropriate context to topics touched in this ms.

I have no issues with this whatsoever.....

Experimental design

appropriate and effective

Validity of the findings

The findings are all valid and well supported.

Additional comments

n/a

·

Basic reporting

This paper presents the results of a valuable experiment that tests whether settlement-stage oyster larvae are influenced by the addition of acoustic playback using recordings of oyster reefs. The study finds significant evidence that oyster larvae can detect and orient towards settlement sites with playback, particularly earlier in the spawning season when density of larvae is lower.

There are a few missing details (see general comments) that it would be useful to include, and I have some suggestions for rewordings, but generally the paper is well written and presented.

Experimental design

The experiment is well designed with multiple sound files used and counterbalanced replication.

Validity of the findings

The findings are valid and clearly presented.

Additional comments

Major comments

Although published in detail in earlier papers, it would be helpful to provide some information on the acoustic conditions of the original oyster reefs from which the recordings were taken.

Also, to qualify the use of playback it would be helpful in Fig 2 to include spectral analysis of the original recordings as well analysis of the playback conditions.

More detail is needed on the equipment used for making the recordings, the sound systems used for playing recordings back in the field, and the methods used to derive spectral analyses of acoustic conditions (e.g. bin width, FFT length, duration of recording analysed, example or summary of several recordings?)

Currently there is no information on the potential hearing mechanisms of the oyster larvae. While this may be speculative, it would still be a useful inclusion that could direct future research.

This is only one experiment, with 8 trials. Check throughout the paper for consistency.


Minor comments

Abstract

Suggest changing 4th sentence to start: Here we show in a field experiment that the…

Remove “Measured”

Explain the term “collectors”


Introduction

Line 44: significant challenge TO MARINE LARVAE.

L48: Change “signal” to “aid”

L49: Change “advantageous” to “suitable”

L60: environment AT a particular…

L65: Lillis et al 2014a should come first (not 2014b); also L94, L97, L125

L68: avoid superlative: tremendous

L72: shorten to: test whether sounds of oyster reefs could enhance recruitment…

L74: change “valuable” to “important” (value requires quantifying)

L88: remove “aspects of”


Materials and Methods

L109: pluralise: used recordings taken from

L110: Not sure what Fig 2 refers to. Was expecting a map

L111: permit numbers in parentheses

L125: How was “prevention of interference” confirmed?

Fig. 2: What was generating the sound <800 Hz if not playback of oyster reef noise?

L150: Is the formula for an Exact binomial test needed, or is it standard fare?


Results

L166: these are not 8 separate experiments, rather 8 trials of a single experiment

L168: simplify: There was no difference in settlement rates between sound treatments in the last two trials, which were conducted during a period of high larval settlement.

Fig 3: Use same colours for figs A & B

Table 1: Give year as well as day/month


Discussion

L187: change to: selects substrate using acoustic cues

L202: change to: integrates patterns

L210; manipulation WAS predicted

L213: remove “short-term”

L213: change to: additional sound could lead to long-term differences in oyster density

L215: recruitment correlates with

L222: remove “known to be”

L223: larval settlement occurs with less selectivity

L226: patterns do not relate to acoustic conditions when


References

Italicise species names

Lowercase titles except first word

Capitalise first letters of journal titles

---

## Round 0.2 · accepted · Accept

Dear Authors,

Thank you for submitting your corrected manuscript. I am pleased to inform you that your manuscript has now been accepted for publication and thank you for submitting to PeerJ.